# The miRNA Contribution in Adipocyte Maturation

**DOI:** 10.3390/ncrna10030035

**Published:** 2024-06-12

**Authors:** Alessandro Giammona, Simone Di Franco, Alessia Lo Dico, Giorgio Stassi

**Affiliations:** 1Institute of Molecular Bioimaging and Physiology (IBFM), National Research Council (CNR), 20054 Segrate, Italy; alessia.lodico@ibfm.cnr.it; 2National Biodiversity Future Center (NBFC), 90133 Palermo, Italy; 3Laboratory of Cellular and Molecular Pathophysiology, Department of Precision Medicine in Medical, Surgical and Critical Care (Me.Pre.C.C.), University of Palermo, 90127 Palermo, Italy; giorgio.stassi@unipa.it

**Keywords:** mesenchymal stem cells (MSC), adipose tissue, miRNA, regenerative medicine, adipose stem cells

## Abstract

Mesenchymal stem cells, due to their multipotent ability, are considered one of the best candidates to be used in regenerative medicine. To date, the most used source is represented by the bone marrow, despite the limited number of cells and the painful/invasive procedure for collection. Therefore, the scientific community has investigated many alternative sources for the collection of mesenchymal stem cells, with the adipose tissue representing the best option, given the abundance of mesenchymal stem cells and the easy access. Although adipose mesenchymal stem cells have recently been investigated for their multipotency, the molecular mechanisms underlying their adipogenic potential are still unclear. In this scenario, this communication is aimed at defining the role of miRNAs in adipogenic potential of adipose-derived mesenchymal stem cells via real-time PCR. Even if preliminary, our data show that cell culture conditions affect the expression of specific miRNA involved in the adipogenic potential of mesenchymal stem cells. The in vitro/in vivo validation of these results could pave the way for novel therapeutic strategies in the field of regenerative medicine. In conclusion, our research highlights how specific cell culture conditions can modulate the adipogenic potential of adipose mesenchymal stem cells through the regulation of specific miRNAs.

## 1. Introduction

Adipose tissue is an endocrine organ that acts as a reservoir for energy triglycerides, and in case of necessity, it mobilizes energy by releasing fatty acids. This tissue is managed by endocrine and metabolic responses, as well as cellular composition, through the potentially toxic build-up of lipids and their mobilization to limit biotoxicity due to lipid excess in peripheral organs [1,2]. Adipose tissue plays a crucial role in buffering chronic overnutrition by storing excess energy in the form of lipids. However, if this energy storage capacity is exceeded, it can result in pathological accumulation of lipids in other organs. There are three main types of adipose tissues, including subcutaneous and visceral white anabolic white (WAT), catabolic brown adipose tissue (BAT), and mixed adipose tissue. White adipose tissue mainly controls energy excesses by promoting hypertrophy or hyperplasia, which increases its energy-buffering ability. To achieve this, the tissue uses committed progenitors. WAT has been historically defined by anatomical location and the presence of parenchymal cells containing a single large lipid droplet. Researchers have debated the high adipocyte plasticity [3].

It is crucial to conduct an in-depth study of all the cellular types that make up adipose tissue, as well as the external factors that influence their development and adaptability. Adipose tissue is highly heterogeneous, with most cells exhibiting significant multipotency. The tissue primarily consists of mature adipocytes, in addition to a mixture of small mesenchymal stem cells (MSCs) and pre-adipocytes, endothelial cells, macrophages, and T regulatory cells. Pre-adipocytes can proliferate and differentiate into mature adipocytes, providing the adipose tissue with a constant and very high functional adaptability (Figure 1) [3,4]. The process of adipose tissue formation is well-regulated and occurs in different stages, with varying cell organization and differentiation pathways. Several studies have shown that different adipose tissue depots form at different times and possess unique molecular characteristics, indicating regional differences between them. Additionally, research into stem cell adipose biology has focused on understanding the molecular characteristics and differentiation potential of progenitor cells within this tissue [5,6,7,8]. Stem cells reside in specialized locations known as stem cell niches, where they remain dormant until adjacent cells or external signals from the microenvironment stimulate them to proliferate and differentiate. Adipogenesis is closely linked to other developmental processes, with angiogenesis being the most important among them [9,10]. Moreover, this process directly influences pre-adipocyte proliferation and differentiation, thus reducing vascularization during obesity and inducing apoptosis in dysfunctional adipocytes [11,12,13].

Furthermore, adipose tissue stromal components provide structural support and biochemical signals, maintaining tissue functionality. Extracellular matrix (ECM) components impact the mesenchymal lineage fate, proliferation, and cell differentiation, while mesenchymal progenitors play an essential role in matrix remodeling [14,15,16]. Adipocyte maturation upregulates collagen IV expression and several laminin complexes, while it reduces fibronectin expression during adipocyte differentiation [17,18]. Another regulation of adipogenesis regarding the so-called adipocytokines, this secreted class of molecules exerts a significant effect on every adjacent cell.

Accordingly, it is widely known that the adipose tissue serves not only as an energy storage system, but also as an important endocrine and immune organ, and still, all whole-tissue functions are not fully characterized.

Therefore, the adipocyte differentiation process mainly comprises two stages: the “determination phase” and then the “terminal differentiation phase” [19,20].

During the beginning phase of adipogenesis, C/EBP-β and C/EBP-δ factors are identified as crucial transcriptional factors. These factors accumulate, causing adipose cells to re-enter the cell cycle and promoting the transition from active G1 to S phase. Specifically, when C/EBP-β is hyperphosphorylated and then activated by glycogen synthase kinase-3 B and mitogen-activated protein kinase (MAPK), it works together with C/EBP-δ to stimulate the expression of PPARγ and C/EBP-α. Hence, both C/EBP-β and C/EBP-δ play a vital role in regulating the transcription of genes that are critical for adipocyte differentiation [21,22]. Therefore, both factors promote their expressions, but meanwhile, they can promote many other genes whose expression could induce specific adipocyte commitment or fate [23,24].

PPARγ represents an important nuclear receptor, and it is considered a crucial transcription factor that drives brown or white adipocyte differentiation. Two PPARγ isoforms have been described: PPARγ1, which is constitutively expressed, and the expression of which is characteristic of fat tissues; and PPARγ2, regulated by the previous one, which regulates mainly adipocyte differentiation [25,26]. Adipocyte commitment results from the equilibrium between these factors, due to the dual face of these factors, which could be considered pro- and anti-adipogenicity transcription factors.

The Kruppel-like factors (KLFs), instead, of C2H2 a zinc-finger factor, regulates adipose tissue apoptosis, proliferation, and differentiation. There are different isoforms of this gene with different biological and molecular roles, such as KLF15 (which promotes GLUT-4 expression), KLF5 (induced by C/EBP-β and C/EBP-δ in the early adipocyte maturation), KLF9 (key pro-adipogenicity transcription factor through the middle adipogenesis stage), KLF2 and KLF7 (anti-adipogenicity factors, and in particular, KLF2 represses the Pparg2 promoter) [27,28,29].

Many other groups have studied other transcription factors that could act specifically on the repression of adipogenesis, such as the GATA-binding family or Forkhead Box O1 and A2 (FOXO1 and FOXA2) families. Thus, we could conclude that adipogenesis results from positive or negative stimuli and regulation factors, many of which are hormones, cytokines, growth factors, and some pharmacological compounds [30,31,32,33,34].

Adipose tissue shows a complex and dynamic mixture of cellular and non-cellular elements, including progenitors and blood–lymphatic vessels recruited by immune cells, nerves, and extracellular matrix elements, but pre-adipocytes play a crucial role in adipose tissue plasticity. In particular, some studies regarding pre-adipocyte miRNA expression have shown significant alterations during adipose tissue development and in obesity, and as a consequence, a deeper insight into miRNAs’ role during the different proliferation and differentiation phases of adipocyte commitment or maturation could provide innovative and efficient regenerative medicine strategies. miRNAs are one of the most important biology and medicine scientific discoveries. miRNAs consist of a short class of RNA molecules, which are about 20 nucleotides long, endogenous, single-stranded, and non-coding. They are involved in the negative post-transcription gene expression pairing of some specific mRNAs 3′UTR. The miRNA mediates RNA degradation or its translational repression [35]. Scientific evidence shows that miRNAs play a crucial role in the regulation of several biological behaviors, such as embryonic maturity, metabolic programs, and cell death [36]. Currently, thousands of miRNA gene codes have been recognized [37]. The interaction involves the formation of a double-stranded assembly between the miRNA seed and the mRNA “target” [38,39].

Kim et al. 2010 [40] have described the relationship between miRNAs and the regulation of early adipocyte differentiation. They studied how miR-27b overexpression correlates with adipogenesis, and both groups found that its abundance during human adipocyte differentiation decreases the PPARδ and C/EBPα induction. Therefore, the miR-27 gene family is potentially an important class of negative adipogenic regulators, useful as an anti-adipogenic factor [31,41]. Then, another miRNA, the miR-519d, was studied by Martinelli et al. 2010 [42], and was found to be crucial in adipocyte development. This specific miRNA suppresses the translation of PPARα protein in a dose-dependent manner, increasing lipid accumulation during preadipocyte differentiation. Moreover, Yang et al. [34] described the implication of miR-138 expression in adipogenicity differentiation. MiR-138 is downregulated during the adipogenicity differentiation of human adipose tissue, while its overexpression in mesenchymal stem cells reduces lipid droplet accumulation. In addition, Sun et al. 2009 [43] highlighted the miR-31 role in MSCs. It was observed that once miR-31 was upregulated, the MSC adipogenic differentiation was repressed, and this was driven by C/EBPα expression. In addition, Tang et al. 2009 [22] reported that, during the differentiation of adipose-derived stem cells (ADSCs), the expression of miR-31 and miR-326 were significantly down-regulated. Furthermore, Esau et al. 2004 [30] showed that miR-143 levels increased in differentiating adipocytes, and its ablation inhibited adipocyte differentiation, acting on the MAPK signaling pathway, even if in the terminal differentiation step. Moreover, ERK5, which is involved in adipocyte differentiation, represents a target gene of miR-143, and Oskowitz et al. 2008 [32] observed that relative overexpression promoted adipogenicity differentiation. Instead, in 2011, Ling et al. [44] studied the miR-375, which suppresses the phosphorylation levels of ERK1/2 and consequently promotes adipocyte differentiation [45,46]. Therefore, miRNA inhibitors toward MAPK could be used as a novel approach to reduce adipocyte differentiation and decrease lipid accumulation. In addition, the miR-143 negatively affects glucose homeostasis through the activation of the Akt pathway and specifically downregulates the oxysterol-binding-protein-related protein 8 (ORP8) [45]. Yi et al. 2011 [47] described how the miR-143 could enhance adipogenesis with pleiotrophin (PTN) silencing, resulting in a negative adipogenesis differentiation through PTN/PI3K/AKT pathway. Despite the miR-375 modulates, by targeting PDK, the glucose-mediated stimulatory effect on insulin gene expression, inactivates the Akt pathway thanks to phosphatidyl-inositides generated by PI3K [45]. Other miRNAs, including miR-210, miR-148a, miR-194, and miR-322, which induce adipogenesis, repressing Wingless-type MMTV integration site family members (Wnt) signaling, have been described. Other miRNAs, such as miR-344, miR-27, and miR-181, inhibit adipogenesis by secreting glycoproteins through their Frizzled (Fz) receptors and low-density lipoprotein receptor-related protein (LRP) co-receptors. As described in the literature, Wnt signaling blocks adipocyte differentiation, inhibiting PPARγ and CEBPα expression [48,49]. Furthermore, the relationship between lipid metabolism and miRNA modulation has been highlighted in several papers. For instance, miR-210 overexpression induces hypertrophy and lipid droplet formation in fat cells, and its inhibition promotes the adipogenesis block [50]. miR-103 is reported to be upregulated during differentiation of human pre-adipocytes, and its overexpression in the presence of adipogenic stimuli increases triglyceride accumulation and adipogenic gene expression [43,51,52,53,54]. In mature adipocytes, long terminal differentiation and the upregulation or downregulation of specific miRNAs, such as miR-221, miR-125b, miR-34a, and miR-100, have different effects. MiR-34a, positively upregulated during adipogenesis, is associated with increase in BMI [30]. Despite miR-448 suppressing adipocyte differentiation, Kruppel-like factor 5 (KLF5) contains a selective miR-448 binding site. Overexpression of miR-448 in pre-adipocytes suppresses KLF5, triglyceride accumulation, and adipogenic gene expression [31]. Moreover, miR-15a inhibition reduces preadipocyte size while promoting adipocyte proliferation. In preadipocytes, miR-15a has been shown to target DLK1 at the mRNA and protein level [55]. Furthermore, miR-222 and miR-221 are decreased during adipogenesis but upregulated in obese adipocytes; contrarily, miR-185 is upregulated in mature adipocytes while downregulated in obese men. In summary, there is a strong influence of the miRNAome on potential differentiation, and several miRNAs have been identified which can accelerate or inhibit adipocyte turnover (Table 1 and Figure 2).

## 2. Results

### 2.1. Different Culture Conditions Affect miRNA Expression

The miRNAs analysis comparison was performed between the two adipose stem subpopulations, ASphCs and ADSCs (commercial adipose cells grown in adhesion), and their relative mature adipocyte populations, ADA (ADSC-derived adipocytes) and SDA (ASphCs-derived adipocytes). The lists of miRNAs and the obtained cluster-grams highlighted a wide negative transcriptional regulation when we compared ASphCs, ADSCs, and ADA with SDA (Figure 3A). Furthermore, when we compared the two adipose stem populations, it was possible to note a different miRNA pattern of expression with a score index of 85% (Figure 3B). Thus, we could assume that the culture conditions, like non-adherence and absence of fetal bovine serum (FBS), are important for the expression of those miRNA expressions that are mutually expressed in ASphCs or ADSCs. Indeed, we demonstrated that ASphCs, which grow in the presence of a specific stem FBS-free medium, reside in a quiescent state and possess greater cellular plasticity than ADSCs [57,58]. Therefore, these multipotential differences could be due to these specific miRNA expressions that are involved in the key biological processes.

In more detail, ASphCs are endowed with higher expression levels of miR-126, which is a negative cell growth and angiogenesis regulator (silencing VEGF, FGF, and EGF). ASphCs also show overexpression of the miR-146 family, which is composed of mir146a and mir146b, which promote adipocyte commitment. In addition, ASphCs show high expression levels of miR-25, which is a p27 inhibitor control. ADSCs, contrarily, have an increased expression of miR-143 and a negative adipocyte commitment regulator, reported as an adipogenesis induced through the MAPK signaling pathway and pleiotrophin (PTN) silencing. Moreover, miR-143 impairs glucose homeostasis through the Akt pathway and downregulates the oxysterol-binding protein-related protein 8 (ORP8). We also observed a higher level of miR-145, which inhibits SOX9 and ROCK genes, regulating mesenchymal stem cell differentiation toward angiogenesis, as well as miR-494 overexpression, which inhibits both the growth and angiogenesis potential of mesenchymal stem cells [42,47,48,49] (Figure 3B and Table 2).

### 2.2. MicroRNAs Regulate Adipocyte Differentiation

After completing the comparison of mesenchymal stem cell cultures, we conducted a thorough investigation of the difference in miRNA regulation between mesenchymal stem cell cultures and their derivatives and mature adipocytes (Figure 4A,B). Our analysis revealed that adipogenesis is a process that is well-regulated by miRNA families, as highlighted by the huge number of miRNAs that were overexpressed. In particular, both the adipose stem cell cultures (ASphC and ADSCs) showed a higher expression of these specific miRNAs: miR-100, miR-10a, miR-143, miR-197, miR-222, miR-410, miR-484, and miR-31. Specifically, miR-31 represses adipogenesis, inhibiting the C/EBP gene. miR-10a suppresses adipose inflammation through the TGF-β1/Smad3 signaling pathway, regulates the preadipocyte proliferation and differentiation through KLF11 inhibition, and restrains adipogenic differentiation, targeting Map2k6. Instead, both miR-100 and miR-197 are associated with obesity, but a high level of miR-100 correlates with the block of cell cycle progression by inhibiting the mTOR pathway, thus leading to stemness maintenance at the expense of adipocyte commitment. In addition, miR-222 is upregulated in obese individuals and inhibits adipogenesis by targeting PPARg and CEBPa. Moreover, the high expression of miR-410 and miR-484 correlates with a lack of preadipocyte differentiation, first by targeting IRS-1 and second by inhibiting SFRP1 or γ-Synuclein (SNCG) expression. This silencing causes the impairing of the metabolic functions in fat cells and therefore allows adipocyte differentiation and adipose tissue enlargement. Our analysis revealed that ASphCs mutually showed a higher expression of miR-27, miR-125b, and miR-186 compared to SDA. These miRNAs are crucial for adipogenesis and are defined as antiadipogenic miRNAs. In particular, miR-27 decreases PPARδ and C/EBPα induction, and their inhibition critically prevents adipocyte maturation. Furthermore, overexpression of miR-27 and high levels of miR-186 activate the Wnt pathway, which may explain the undifferentiated status of these stem cell populations. Additionally, the adipocyte lineage commitment of ASphCs is under the regulation control of miR-125b, which is responsible for MAPK silencing [31]. From the comparison between ADSC and ADA, we individuated the mutually higher expressed miRNA as relevant; the mir138’s overexpression in mesenchymal stem cells reduces lipid droplet accumulation and usually results in downregulation during adipogenicity differentiation [34]. Our analysis revealed a significant lack of miRNA expression in the mature population compared to the stem subpopulations. Specifically, only nine families of miRNAs were exclusively expressed, and surprisingly, only two miRNAs showed higher expression in both the mature populations compared to the stem subpopulations. All this information is further illustrated in Table 2 and Table 3.

A possible reason for this poor miRNA expression can be attributed to the comparison with greater expression of miRNA of the stem counterpart, but also to the limited nuclear component of the differentiated counterpart and the difficulties of manipulating biological samples so devoid of nucleic material.

Thus, we conducted a more in-depth analysis to compare the collected literature references concerning pro-adipogenic or anti-adipogenic miRNAs and our results. This analysis allowed us to identify nine crucial miRNAs to deeply investigated in order to uncover the crucial biological adipogenic mechanisms from stem cell subpopulation and their mature counterparts (Figure 5 and Table 4).

## 3. Discussion and Conclusions

As described in many scientific works, adipose tissue is often used in regenerative medicine as an autologous filler for the correction of morpho-functional defects. In particular, adipose-derived mesenchymal stem cells are increasingly recognized for their potential in regenerative medicine due to several key advantages. Unlike bone marrow, which has traditionally been the primary source of mesenchymal stem cells but involves a painful and invasive collection process with limited cell yield, adipose tissue offers a more abundant and easily accessible source of these cells. Adipose-derived mesenchymal stem cells have demonstrated significant promise in various therapeutic applications owing to their ability to differentiate into multiple cell types, including bone, cartilage, and fat cells.

Thus, a better understanding of the molecular mechanisms underlying the adipogenic potential of mesenchymal stem cells could help in the optimization of the collection, culture, and treatment of this cell subset to improve their regenerative capacity. In this scenario, microRNAs (miRNAs) can be considered as crucial drivers of adipogenic commitment of adipose-derived mesenchymal stem cells, and at the same time as promising targets to increase the clinical impact of regenerative medicine procedure based on the use of this cell subset.

Our data, obtained from this vast analysis of miRNAs and adipose stem cell subpopulations together with their differentiated progenies, suggest that adipogenesis is carefully controlled by several factors (passage of cells, cell culture medium, adhesion versus ultra-low adhesion) and at many different levels. All these variables directly affect the expression of specific miRNAs involved in the adipogenic potential of adipose-derived mesenchymal stem cells. We can speculate on a possible viable engineering therapy where an ectopic expression of miRNAs could be exploited to promote a specific commitment (i.e., we could obtain a pre-adipocyte population compared to a population of mature adipocytes overexpressing mir27a or the mir31, limiting the harmful effect of adipocytes in lipofilling in cancer patients subjected to demolition surgery). Alternatively, we could try to overexpress mir100 or mir146a, succeeding in limiting the onset of breast cancer or in reverting a more aggressive phenotype of breast cancer to one that responds to anti-cancer therapies. Moreover, in the case of ectopic mir455, we could induce an adipose-type switch from WAT to BAT and thus minimize the damaging effects of obesity and paracrine secretion of harmful white adipose tissue [50,54]. Moreover, another strategy could rely on the use of microRNAs identified as possible candidates for the modulation of adipocyte-based processes. The delivery of such microRNAs through nanovesicles could deepen our research and mechanistic understanding of the biological and differentiation processes shown.

In summary, even based on preliminary transcriptomic data, our results, if validated at a pre-clinical level by performing in vitro/in vivo analyses, could help to better define the optimal culture conditions and the main molecular driver of adipogenic differentiation, with important clinical impact in regenerative medicine.

## 4. Materials and Methods

### 4.1. Adipose Tissue Samples and Cell Culture

The adipose tissues were extracted from a lipoaspirate tissue biopsy, in compliance with our department’s policy. Patients were treated in compliance with our department’s policy, following the patient’s written consent on adipose tissue harvesting and its use for research purposes. The study was approved by the ethics committee Palermo-1 Polyclinic Paolo Giaccone of Palermo, with the report N°1/2016.

Adipose tissue sampling was performed via local infiltration of 150 cm^3^ of Klein solution under conscious sedation. A cannula (10-gauge diameter) connected to a lock-type attack syringe was used for the liposuction, and a mean of 100 cm^3^ subcutaneous adipose tissue was extracted from each patient. The samples were centrifuged at 700× *g* for 5 min, and the bottom layer, composed of MSCs and blood elements, was collected and digested for 30 min at 37 °C in the presence of collagenase (1.5 mg/mL; Thermofisher Scientific, Waltham, MA, USA) and hyaluronidase (20 mg/mL; Merck, Darmstadt, Germany). The digested samples were centrifuged and washed with PBS. The obtained pellet was suspended in serum-free, stem-cell-specific medium supplemented with bFGF (10 ng/mL; Merck, Darmstadt, Germany) and EGF (20 ng/mL; Merck, Darmstadt, Germany) in ultra-low adhesion culture flasks (Corning, Corning, NY, USA), as described previously [57]. In these conditions, the cells grew as floating spheroids called AD-MSCs StemPro™ human adipose-derived stem cells (ADSCs; Thermofisher Scientific, Waltham, MA, USA). They were cultured as the manufacturer recommended and used as the MSC control.

### 4.2. Adipogenic Differentiation

Both primary adipose stromal cells grown as floating spheroids (ASphCs) and ADSCs were trypsinized for 5 min at 37 °C and cultured into adherent 24-well cell culture plates (50.000 cells/well). Cells were cultured in a STEMPRO^®^ Adipogenesis Differentiation Kit (Thermofisher Scientific, Waltham, MA, USA) for up to 28 days. Cell viability, adhesion, and differentiation were assessed by daily observation using optical microscopy. We assessed the mature adipocytes with AdipoRed TM Assay Reagent (Lonza, Basilea, Switzerland). We referred to “SDA” to describe ASphC-derived adipocytes, and to “ADA” as ADSC-derived adipocytes.

### 4.3. miRNA Gene Expression

We extracted the total RNAs using TRIzol^®^ Reagent (Thermofisher Scientific, Waltham, MA, USA), following the manufacturer’s instructions. We evaluated miRNA expression with the Megaplex pools protocol specific for a set of 384 microRNAs (pool A), as recommended by the manufacturer (Thermofisher Scientific, Waltham, MA, USA). The relative quantification of microRNA expression was calculated using the equivalent Ct values, where the original CT values were projected to 100% target efficiency. Clustergrams and scatter plots were generated using the RT2 Profiler PCR Array Data Analysis v3.5 software (Qiagen, Hilden, Germany).

All the experiments were performed in triplicate and normalized using the global normalization method. microRNA assays were performed for ASphCs, ADSCs, and their differentiated cells. miRNAs with 3-fold changes were considered for analysis.

## Figures and Tables

**Figure 1 ncrna-10-00035-f001:**
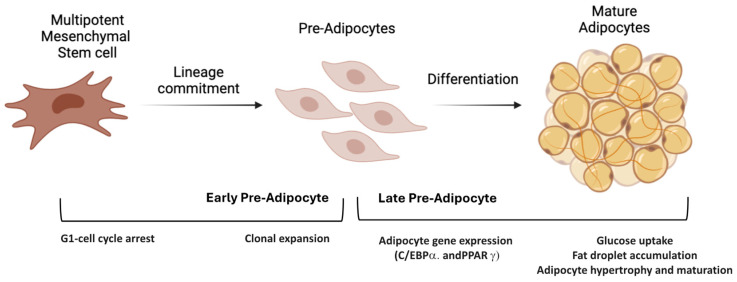
Mesenchymal stem cell commitment. The differentiation of mesenchymal stem cells into mature adipocytes involves both a preliminary lineage commitment and subsequent terminal differentiation.

**Figure 2 ncrna-10-00035-f002:**
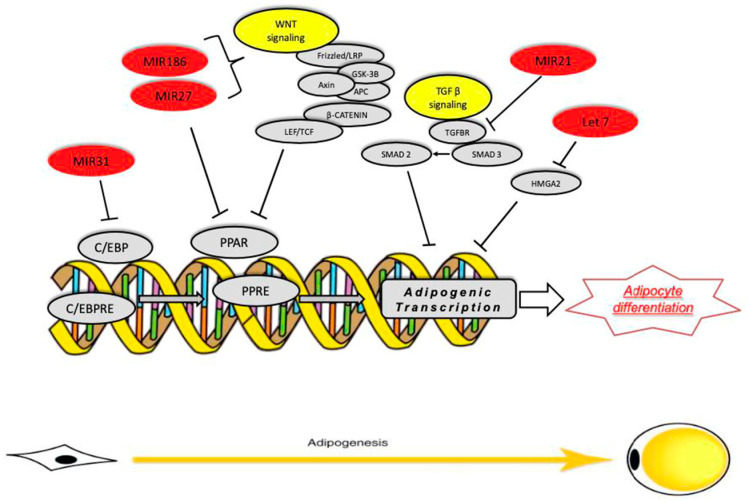
miRNA families that regulate specific target genes involved in adipogenesis.

**Figure 3 ncrna-10-00035-f003:**
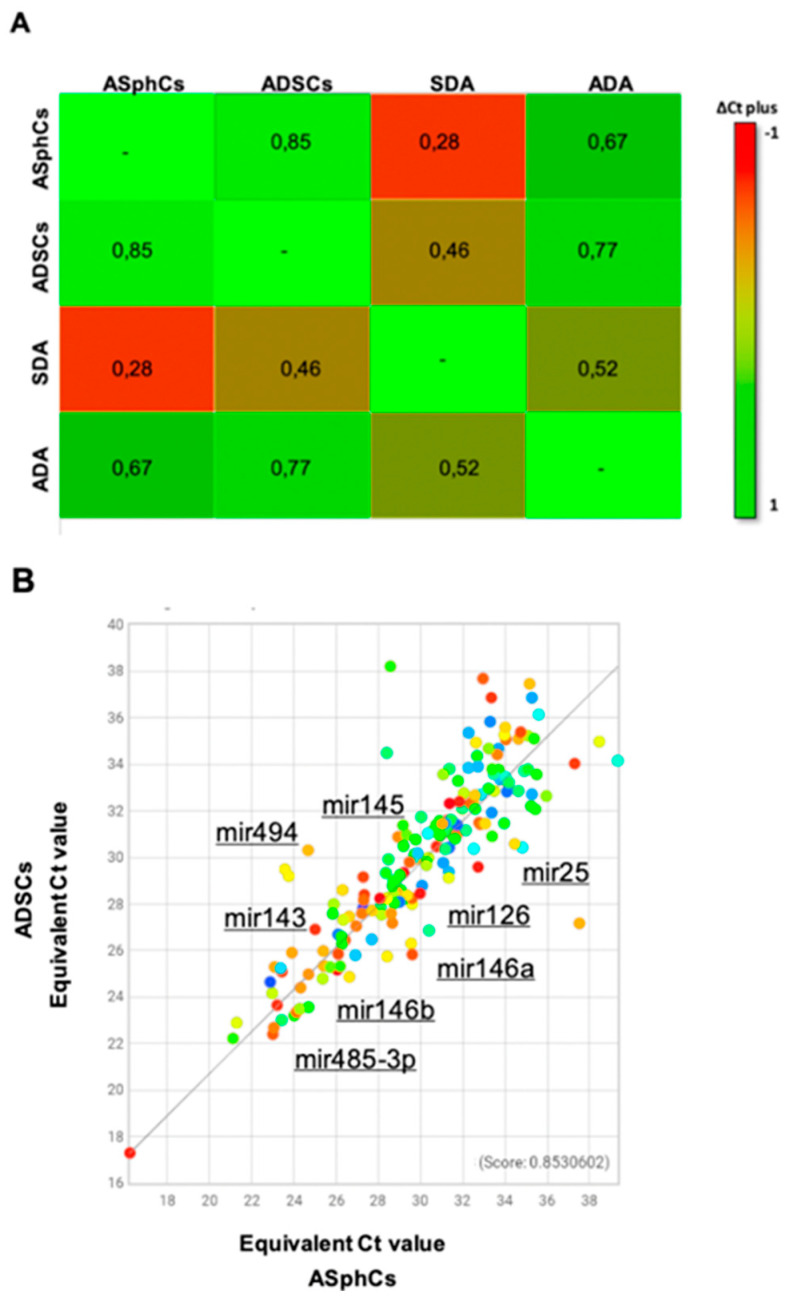
The differences in terms of multi-difference efficiency could be due to miRNA expression. (**A**) Clustergrams of mesenchymal stem cell and their derived mature adipocytes, ADA (ADSC-derived adipocytes) and SDA (ASphC-derived adipocytes). The score index is expressed using equivalent Cт values, where the original Cт values are projected to 100% target efficiency. (**B**) Scatter plot of human miRNA expression, and miRNA expression reported as fold change ADSCs compared with ASphCs.

**Figure 4 ncrna-10-00035-f004:**
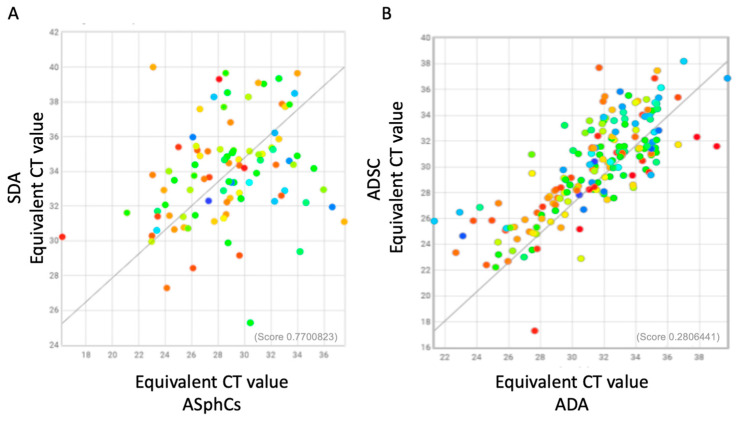
The miRNAs acting on different pathways. (**A**) Scatter plot of human microRNA expressed by mesenchymal stem cell ASphCs and their derived mature adipocytes, SDA (ASphC-derived adipocytes). (**B**) Scatter plots of human microRNA expressed by mesenchymal stem cell ADSC and their derived mature adipocytes, ADA (ASphC-derived adipocytes).

**Figure 5 ncrna-10-00035-f005:**
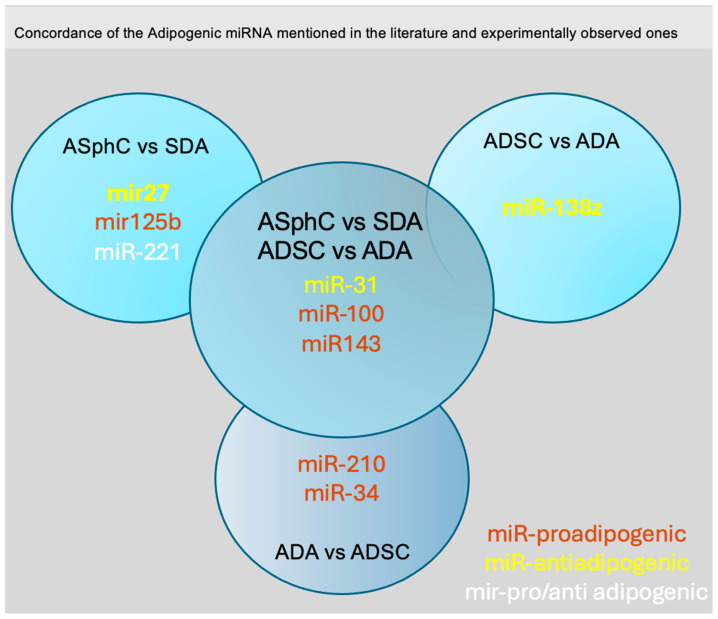
Graphic synthesis of comparative analysis between the miRNAs described in the literature and the miRNAs observed from experimental data.

**Table 1 ncrna-10-00035-t001:** Summary of described microRNAs involved in the regulation of adipogenesis.

miRNA	Reported Mechanisms	Ref.
miR-27b	Decreases adipogenesis impairing PPARδ and C/EBPα induction	[31]
miR-519d	Suppresses, in a dose-dependent way, the translation of PPARα protein and increases lipid accumulation during preadipocyte differentiation	[42]
miR-138	Downregulated during the adipogenicity differentiation, but its overexpression in mesenchymal stem cells reduces lipid droplet accumulation	[34]
miR-31	Represses adipogenesis and is driven by C/EBPα expression	[22,43]
miR-326	Down-regulated long adipose-derived stem cell differentiation	
miR-143	Promotes adipogenesis through the MAPK signaling pathway, and by silencing pleiotrophin (PTN). Impairs glucose homeostasis via Akt pathway and ORP8	[30,32,56]
miR-210, miR-148a, miR-194, miR-322	Promotes adipogenesis, repressing Wingless-type MMTV integration site family members (Wnt)	[49]
miR-344, miR-27	Impairs adipogenesis through Frizzled (Fz) receptors and (LRP) co-receptors	[49]
miR-375	Promotes adipocyte differentiation, suppressing ERK1/2 Modulates the glucose stimulatory effect on insulin gene expression by targeting PDK	[44]
miR-210	Induces hypertrophy and lipid droplet formation; its inhibition promotes adipogenesis arrest	[49]
miR-103	Upregulated during pre-adipocytes differentiation increases triglyceride accumulation and adipogenic gene expression	[50]
miR-125b, miR-34a, miR-100	Upregulated during adipogenesis and is associated with high BMI	[31]
miR-448s	Suppresses adipogenesis through suppression of Kruppel-like factor 5 and triglyceride accumulation	[31]
miR-15a	Its inhibition reduces preadipocyte size while promoting adipocyte proliferation, targeting DLK1	[31]
miR-222, miR-221	Decreased during adipogenesis but upregulated in obese adipocytes	[31]
miR-185	Upregulated in mature adipocytes but downregulated in obese patients	[31]

**Table 2 ncrna-10-00035-t002:** Mesenchymal stem cells and miRNA comparison.

ASphCs vs. ADSCs			
miRNA	Fold Change	Biological Role of miRNA	miRNA Targets
miR-126	5.6	Cell growth and angiogenesis control [59,60]	VEGF, FGF, EGF, PI3K/Akt and MAPK/ERK pathways
miR-146a	5.9	Cell cycle regulation [61,62,63]	ErbB4, PPARγ
miR-146b	5.4	Adipogenesis commitment promotion [64,65,66]	KLF7, PPARγ2 and SOX9
miR-25	4.72	Adipogenesis suppression [38,67,68]	p27, KLF4
miR-145	–2.71	Regulates mesenchymal stem cell differentiation and promotes angiogenesis [69,70,71]	SOX9 and ROCK, TGF-β3, ETS1
miR-143	−3.24	Adipocyte commitment regulator [30,47,56]	AKT, glucose metabolism and ERK5
miR-494	−3.6	Inhibits both the growth and angiogenesis potential of mesenchymal stem cells [41,72,73]	PGC1-α signaling

**Table 3 ncrna-10-00035-t003:** miRNAs commonly more or less expressed by adipose mesenchymal stem cells versus their derived adipocytes.

	ASphCs vs. SDA	ADSCs vs. ADA	
miRNA	Fold Change		Biological Role of miRNA
miR-100	7.3	2.17	Upregulated during adipogenesis, associated with high BMI
miR-10a	4.29	3.1	Modulates adiposity and suppresses inflammation through the TGF-β1/Smad3 signaling pathway. It regulates preadipocyte proliferation and differentiation by targeting KLF1, promoting the cell cycle, and restraining adipogenic differentiation by targeting MAP2k6 and FASN
miR-143	5.39	3.22	Promotes adipogenesis through the MAPK signaling pathway and pleiotrophin (PTN) silencing. It impairs glucose homeostasis through the Akt pathway and downregulation of the oxysterol-binding protein-related protein (ORP8)
miR-197	7.08	3.74	Overexpressed in obese individuals
miR-222	10.5	3.08	Upregulated in obese individuals and inhibits adipogenesis by targeting PPARG and CEBPA
miR-410	6.75	3.21	Inhibits adipocyte differentiation by targeting IRS-1
miR-484	8.39	3.94	miR-484 targets SFRP1 and affects preadipocyte proliferation, differentiation, and apoptosis. MiR-424(322)/503 targets γ-Synuclein (SNCG), a factor that controls metabolic functions in fat cells, allowing adipocyte differentiation and adipose tissue enlargement to occur.
miR-31	12	3.29	Down-regulated long adipose-derived stem cell differentiation; represses adipogenesis; driven by C/EBPα expression
miR-494	−5.11	−2.57	miR-494-3p controls white adipose thermogenesis. Further gain and loss of function studies of miR-494-3p in 3T3-L1 adipocytes have shown that overexpressed miR-494-3p inhibits adipocyte browning, mitochondrial biogenesis, and thermogenesis through PGC1-α
miR-202	−3.82	−4.05	miR-202 promotes the differentiation of 3T3-L1 preadipocyte via inhibiting PGC1β expression

**Table 4 ncrna-10-00035-t004:** miRNAs mutually expressed from the mesenchymal stem subpopulation versus the differentiated adipocyte lines.

ASphCs vs. SDA		ADSCs vs. ADA	
miRNA	Fold Change	miRNA	Fold Change
miR-106	9.04	miR-10b	5.11
miR-106b	7.9	miR-138	5.19
miR-125b	5.32	miR-141	4.23
miR-130b	5.15	miR-149	3.89
miR-17	10	miR-15b	4.09
miR-186	9.1	miR-196b	5.45
miR-191	6.48	miR-197	3.74
miR-193A-3p	4.7	miR-212	3.58
miR-193A-5p	7.1	miR-28	3.9
miR-193b	6.03	miR-32-3p	4.21
miR-195	8.05	miR-328	4.13
miR-199A-3p	10.5	miR-342	5.17
miR-19a	6.3	miR-362	3.26
miR-19b	8	miR-370	3.53
miR-218	9.25	miR-382	5.22
miR-221	9.03	miR-410	3.21
miR-224	6.4	miR-424	3.76
miR-24	7.2	miR-431	4.69
miR-26a	8.7	miR-454	4.13
miR-27	4.7	miR-199b	−3.87
miR-29c	5	miR-210	−3.99
miR-30b	7.4	miR-22	−3.39
miR-30c	6.5	miR-25	−5.96
miR-320	8.2	miR-29a	−3.14
miR-324-3p	7.36	miR-34c	−3.79
miR-331	5.2	miR-59	−4.6
miR-365	6.16	miR-618	−3.59
miR-374	6.02		
miR-376c	5.04		
miR-411	4.6		
miR-452	8.4		
miR-486	8.8		
miR-574	7.27		
miR-744	7.75		
miR-483-5p	−4.1		

## Data Availability

No datasets were created during the study. The raw data supporting the conclusions of this article will be made available by the authors upon request.

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
