# Peer review of "The miRNA Contribution in Adipocyte Maturation"

_ncrna, 2024, doi:10.3390/ncrna10030035_

Round 1

Reviewer 1 Report

Comments and Suggestions for Authors

Supposedly, the authors aim to provide an overview on the contribution of miRNAs in the differentiation process of adipocytes, which is of general interest. For this, they employ two different adipose stem sub-populations, which are differentially differentiated (non-adherent, FBS-free or adherent with FBS), and run miRNA expression analysis via commercial microRNA arrays on the four groups. Unfortunately, the whole manuscript is rather shallow and aimless. It already starts with the fact that the abstract states neither the aim of the manuscript nor the results of the study.

I do not want to offend the authors here, but it is not clear for a long time what the purpose of this “communication” is supposed to be and it sometimes seems as if they are just trying to reach at least a certain number of pages. For example, figure 1 is totally superficial and definitely not needed to understand the process. If at all, it might be fused with figure 2 to provide at least some meaningful information. Moreover, the information in figure 3 and table 1 are redundant just like those of figure 4 and table 2. It is not necessary to illustrate the fold changes in two different ways.

On the other hand, I would recommend summarizing all the studies listed in page 4 and 5 and providing some table giving the miRNAs together with their targets, their effect on the differentiation process, potentially the cell or animal model and the respective references. This would add a lot more value to the introduction than the figures. Consequently, it might be discussed which of the mentioned miRNAs were also found in this study and which were not found to be significantly regulated here. However, it is not clear how representative/relevant the results of this study are in the first place, as it was not even stated how many patients samples were taken. The data might be highly patient-specific and “at least two replicates” for the miRNA gene expression analysis is very questionable from a statistical point of view.

Minor comments:

Page 4 line 133, I would consider literature from more than 14 years ago not necessarily “recent”.

In the results section I am not sure whether “overexpression” is the appropriate term since it just refers to the observed fold changes between two groups and not necessarily an excessive expression of a certain miRNA.

Table 1 and 2, why not simply report negative Fold Changes instead of splitting the tables and switching references?

Page 7 line 237 it says “as shown by data in the literature” but no references are given.

Figure 4 B, why not keeping the undifferentiated cell population at the x-axis like in A?

Figure 5 nicely illustrates how relevant miRNA families modulate adipogenesis, which was somewhat promised by the heading of the manuscript. However, even for a short communication this work lacks some depth and should be revised extensively.

Author Response

Supposedly, the authors aim to provide an overview of the contribution of miRNAs in the differentiation

process of adipocytes, which is of general interest. For this, they employ two different adipose stem

sub-populations, differentially differentiated (non-adherent, FBS-free or adherent with FBS),

and run miRNA expression analysis via commercial microRNA arrays on the four groups. Unfortunately,

the whole manuscript is rather shallow and aimless. It already starts with the fact that the abstract states

neither the aim of the manuscript nor the results of the study.

I’d like first to thank the reviewer for their comments and consideration because after his/her comments

we focused on the manuscript's body text and, as suggested, we started to reorganize the whole paper

starting from the abstract. As you can read, we improved the abstract, thus increasing the clarity of the

aim, and anticipating the results and the conclusions.

I do not want to offend the authors here, but it is not clear for a long time what the purpose of this

“communication” is supposed to be and it sometimes seems as if they are just trying to reach at least a

certain number of pages. For example, figure 1 is totally superficial and definitely not needed to

understand the process. If at all, it might be fused with figure 2 to provide at least some meaningful

information. Moreover, the information in figure 3 and table 1 are redundant just like those of figure 4

and table 2. It is not necessary to illustrate the fold changes in two different ways.

We thank the reviewer for the comment, and we agree with it. For this reason, as suggested, we

generated a single figure by merging the information of Figure 2 into the new Figure1. Regarding the

redundant data shown in Figure3/Table1, and Figure4/Table 2, we think that the Table format is more

convenient to show data from our analysis on miRNAs profile expression, which are now shown in Table

2 , Table 3, and Table 4.

On the other hand, I would recommend summarizing all the studies listed in pages 4 and 5 and providing

some tables giving the miRNAs together with their targets, their effect on the differentiation process,

potentially the cell or animal model, and the respective references. This would add a lot more value to

the introduction than the figures. Consequently, it might be discussed which of the mentioned miRNAs

were also found in this study and which were not found to be significantly regulated here. However, it

is not clear how representative/relevant the results of this study are in the first place, as it was not even

stated how many patient samples were taken. The data might be highly patient-specific and “at least

two replicates” for the miRNA gene expression analysis is very questionable from a statistical point of

view.

As suggested, we revised and summarized all the reported literature references in the new Table 1.

Moreover, we added a graphic synthesis of comparative analysis between the miRNAs described in the

literature and the miRNAs observed from experimental data in the new Figure 5.

Minor comments:

Page 4 line 133, I would consider literature from more than 14 years ago not necessarily “recent”. We

agree and so we corrected the sentence

In the results section, I am not sure whether “overexpression” is the appropriate term since it just refers

to the observed fold changes between two groups and not necessarily an excessive expression of a

certain miRNA. As suggested we used a different way to say, as higher expression.

Table 1 and 2, why not simply report negative Fold Changes instead of splitting the tables and switching

references? Thank you for the good suggestion, we collected the info in one single way (as a negative

fold change)

Page 7 line 237 it says “as shown by data in the literature” but no references are given. The reviewer is

right, we add the Ortega reference to that.

Figure 4 B, why not keep the undifferentiated cell population at the x-axis like in A? We agree with this

reviewer, the design of the graph is an artifact introduced by the software, which automatically define

the order of the samples to be shown in the two axes, to graph the sample with higher expression (so

less equivalent Ct values) in the x-axis.

Figure 5 nicely illustrates how relevant miRNA families modulate adipogenesis, which was somewhat

promised by the heading of the manuscript. However, even for a short communication this work lacks

some depth and should be revised extensively. We thank the reviewer for his/her comment. We decided

to leave the figure, which represents the aims of our pilot study, to determine the possible role of specific

miRNAs in adipogenic differentiation of adipose stromal cells. We agree that this work can be considered a preliminary study, and we are now investigating more in-depth the role of the more

significant miRNAs. These data will be included in another manuscript, which will generate a more

complete original article.

Reviewer 2 Report

Comments and Suggestions for Authors

Manuscript A. Giammona et al. describe the authors' comparative study of miRNA expression spectra (profiles) of mesenchymal stem cells (MSC) cultured in adherent (ADSC) and non-adherent (ASphCs, primary adipose stromal cells grown as oating spheroids) conditions, as well as mature adipocytes derived from these cell cultures - ADSC- and ASphC-Derived Adipocytes. The developed problem as well as the data obtained by the authors are of undoubted interest in terms of regulation of adipogenic differentiation of MSC and fulfillment of various physiological functions by them.

Reviewer’s comments

1) Lines 9-25. Abstract should reflect the following information:

Relevance of the study to the problem (it is about the physiologic role and clinical application of MSC; about miRNA involvement in homeostatic and differentiation processes in MSC: a brief and specific presentation); the aim of the study (the authors do not have it); a summary of the results obtained and conclusions drawn (the authors do not have it)

2) Lines 36-45: in the introduction, the authors mention white and brown adipose tissue. How can this be reflected in the further text of the manuscript, - in the results and discussion!?

3) The description of adipogenic differentiation factors (Lines 83-109) given in the Introduction should be compared more specifically with the description of miRNAs (lines 115-187) involved in the regulation of this process. This will allow the authors to approach the formulation of the aim of the study - to elucidate the possible role of miRNAs in realizing the functions of MSC, adipocytes, as well as other cell types mentioned by the authors. At the end of the introduction, a brief description of the planned experiments and their conclusions should be represented.

4) Fig. 2 raises a number of questions.

-What is the relationship between Growth arrest and clonal expansion?

-It is desirable to give separately the cellular processes involved in adipogenic differentiation (cell division etc) and the factors involved in these processes (C/EBP, PPAR etc.)

-Figs 1 and 2 should be combined into one figure to create a complete picture of the described processes.

5) Figs 3c and 4c-d are redundant and unnecessary, since the information presented therein is already contained in Tables 1 and 2. In Tables 1 and 2, it is desirable (in the form of an additional column) to provide information on (possible) functions/targets of miRNAs: in this case, it will be clear what the authors mean when commenting on the data obtained. In addition, it is highly desirable to present data with a scatterplot (M+/-D) based on the number of replications.

6) In Materials and Methods, information about the construction of clastergrams should be entered.

7) Because some phrases are poorly constructed, authors are strongly recommended to use help in improvement of English.

Minor comments, some misprints mentioned in the text.

1) Lines 51, 86, 88, 93, 100 etc.: highlighting words and sentences in the text is usually used for headings; the importance of individual terms should be explained logically, based on the logic of the presentation of the material

2) Line 63   among them. [10-13]

 3) Line 101   it described

 4) Line 108    forehead families

 5) Fig 2 – growth arrest

 6) Line 138 – sis-negative regulators - ?

Comments on the Quality of English Language

The authors would like to improve the style and correct typos

Author Response

Manuscript A. Giammona et al. describe the authors' comparative study of miRNA expression spectra (profiles) of mesenchymal stem cells (MSC) cultured in adherent (ADSC) and non-adherent (ASphCs, primary adipose stromal cells grown as floating spheroids) conditions, as well as mature adipocytes derived from these cell cultures - ADSC- and ASphC-Derived Adipocytes. The developed problem as well as the data obtained by the authors are of undoubted interest in terms of regulation of adipogenic differentiation of MSC and fulfillment of various physiological functions by them.

Reviewer’s comments

1) Lines 9-25. The abstract should reflect the following information:

Relevance of the study to the problem (it is about the physiologic role and clinical application of MSC; about miRNA involvement in homeostatic and differentiation processes in MSC: a brief and specific presentation); the aim of the study (the authors do not have it); a summary of the results obtained and conclusions drawn (the authors do not have it)

We would like to express my gratitude to the reviewer for the insightful comments and careful consideration. Following his/her feedback, we concentrated on refining the manuscript's body text. As suggested, we commenced a comprehensive reorganization of the entire paper, beginning with the abstract. As a result, we have enhanced the abstract, thereby improving the clarity of our aim (which is represented by a preliminary study to define the role of specific miRNAs in adipogenic differentiation, for multiple clinical applications) and providing a succinct preview of the results and conclusions.

2) Lines 36-45: in the introduction, the authors mention white and brown adipose tissue. How can this be reflected in the further text of the manuscript, - in the results and discussion!?

We agree with this reviewer regarding. Thus, we removed this sentence in the revised version of the manuscript.

3) The description of adipogenic differentiation factors (Lines 83-109) given in the Introduction should be compared more specifically with the description of miRNAs (lines 115-187) involved in the regulation of this process. This will allow the authors to approach the formulation of the aim of the study - to elucidate the possible role of miRNAs in realizing the functions of MSC, adipocytes, as well as other cell types mentioned by the authors. At the end of the introduction, a brief description of the planned experiments and their conclusions should be represented.

4) Fig. 2 raises a number of questions.

-What is the relationship between Growth arrest and clonal expansion?

We agree with this reviewer. For this reason, as suggested, we generated a single figure by merging the information of Figure 2 into the new Figure1, removing the confusing sentence regarding the regulation of proliferative status, which is finely regulated during adipogenic differentiation, and that would deserve more details to be fully appreciated.

-It is desirable to give separately the cellular processes involved in adipogenic differentiation (cell division etc) and the factors involved in these processes (C/EBP, PPAR etc.)

Thanks for the comment. See previous response.

-Figs 1 and 2 should be combined into one figure to create a complete picture of the described processes.

We thank the reviewer’s suggestion and we combined in one figure the information of Figures 1 and 2

5) Figs 3c and 4c-d are redundant and unnecessary since the information presented therein is already contained in Tables 1 and 2. In Tables 1 and 2, it is desirable (in the form of an additional column) to provide information on (possible) functions/targets of miRNAs: in this case, it will be clear what the authors mean when commenting on the data obtained. In addition, it is highly desirable to present data with a scatterplot (M+/-D) based on the number of replications.

We thank the reviewer for the good input. Regarding the redundant data shown in Figure3/Table1, and Figure4/Table 2, we think that the Table format is more convenient to show data from our analysis on miRNAs profile expression, which are now shown in Table 2 , Table 3, and Table 4.

6) In Materials and Methods, information about the construction of clustergrams should be entered. More detail about the generation of the clustergrams have been added in the materials and methods section.

7) Because some phrases are poorly constructed, authors are strongly recommended to use help in the improvement of English. The manuscript has been fully revised, including an extensive English grammar and spell check. We hope the revised version of the manuscript can be now considered suitable for publication.

Minor comments, some misprints were mentioned in the text.

1) Lines 51, 86, 88, 93, 100 etc.: highlighting words and sentences in the text is usually used for headings; the importance of individual terms should be explained logically, based on the logic of the presentation of the material. Thank you for the pointing and correction, we removed the bold form

2) Line 63   among them. [10-13]. We adjusted the text according to the reviewer’s comment.

 3) Line 101   it described Thanks. The text has been adjusted.

 4) Line 108    forehead families. We thank the reviewer for the comment. The text has been corrected.

 5) Fig 2 – growth arrest . thank you for the correction This text has been removed following the merge of Figure1 and Figure 2.

 6) Line 138 – sis-negative regulators - ? We corrected the wrong sentence.

Round 2

Reviewer 1 Report

Comments and Suggestions for Authors

It must have been hard for the authors to receive my rather harsh criticism, but fortunately, they were able to use it to improve the manuscript significantly. In their revised version, the aim of the study is more clearly defined and the overall quality of the data presentation increased. There are still some weaknesses in readability, but introducing smaller paragraphs for distinct aspects and generally revising the language might also be done in the final editing process.

The new figure 1 now contains relevant information and the new table 1 gives a nice overview on the miRNAs that are involved in adipogenesis. I would just recommend harmonizing the structure of the “reported mechanisms” as the authors switch between short bullet points and whole sentences (also, there are some typos). The same holds true for the “miRNA biological role” in table 3. While I like figure 5 and thus would also keep it, I would suggest to put it in the introduction part, as it summarizes miRNA families that are involved in adipogenesis and thus visually supports the given information of the introduction, but it is not related to the outcomes of the study (it doesn´t involve the miRNAs found here).

Moreover, I appreciate that the authors implemented my suggestion by adding a graphic on the shared miRNAs described in the literature and observed from their experimental data. However, based on this I would also suggest to give some details on the specific miRNAs. For example, some of the results here are in contrast to other findings mentioned in the manuscript such as mi-R221 and mi-R138 being upregulated in differentiated cells and mi-R34 being downregulated, while others reported a contrary development in the course of differentiation. Moreover, some of the upregulated miRNAs demonstrate opposing effects, which renders additional studies on their functional relevance important, supporting the first statement of the authors’ discussion. To increase the depth of information in the respective figure one could also color code for pro-adipogenic and anti-adipogenic miRNAs. Generally, I would recommend using a Venn-Diagram and labeling the only two downregulated miRNAs with an arrow or so instead of creating an own “bubble” for them.

While revising the manuscript another question came to my mind. While the authors, report that the different culture conditions of the two investigated adipose cells seem to affect the miRNA expression (results table 2), they did not comment on potential effects on the differentiation and maturation process itself. In the materials and methods section, it is described that differentiation was assessed by daily observation and the AdipoRed TM assay, a statement on the results of that for the two cell sources would be recommended. If there were differences in differentiation efficiency the impact of the different miRNA patterns might be discussed.

Finally, there is still some lack of information. In the section 2.2 (page 8) several references are missing, just as some examples line 254, 257, and 260. Moreover, there are still missing details on the patients such as age and gender and most of all number. What does “At least two replicates were run for each sample” mean for the miRNA Gene expression analysis? “Samples” referring to investigated groups (ASphCs, ADSCs, SDA and ADA) and “replicates” to actual sample size or is there more to it? In case of two patients, how much do the individual results vary?

Author Response

It must have been hard for the authors to receive my rather harsh criticism, but fortunately, they were able to use it to improve the manuscript significantly. In their revised version, the aim of the study is more clearly defined and the overall quality of the data presentation increased. There are still some weaknesses in readability, but introducing smaller paragraphs for distinct aspects and generally revising the language might also be done in the final editing process.

The new Figure 1 now contains relevant information and the new Table 1 gives a nice overview of the miRNAs that are involved in adipogenesis. I would just recommend harmonizing the structure of the “reported mechanisms” as the authors switch between short bullet points and whole sentences (also, there are some typos). The same holds true for the “miRNA biological role” in Table 3. While I like Figure 5 and thus would also keep it, I would suggest putting it in the introduction part, as it summarizes miRNA families that are involved in adipogenesis and thus visually supports the given information of the introduction, but it is not related to the outcomes of the study (it doesn´t involve the miRNAs found here).

We appreciate your recognition of our work on the manuscript and thank you for acknowledging the improvements resulting from your many suggestions. About the new recommendations, we followed them removed the redundancies, and corrected the typos from Table 1, moreover, we followed the other reviewer's indication substituting also in this table all alphabetic author references with numeric author references. In addition, as suggested we moved Figure 5 into the introduction.

Moreover, I appreciate that the authors implemented my suggestion by adding a graphic on the shared miRNAs described in the literature and observed from their experimental data. However, based on this I would also suggest to give some details on the specific miRNAs. For example, some of the results here are in contrast to other findings mentioned in the manuscript such as mi-R221 and mi-R138 being upregulated in differentiated cells and mi-R34 being downregulated, while others reported a contrary development in the course of differentiation. Moreover, some of the upregulated miRNAs demonstrate opposing effects, which renders additional studies on their functional relevance important, supporting the first statement of the authors’ discussion. To increase the depth of information in the respective figure one could also color code for pro-adipogenic and anti-adipogenic miRNAs. Generally, I would recommend using a Venn-Diagram and labeling the only two downregulated miRNAs with an arrow or so instead of creating an own “bubble” for them.

We thank the reviewer for this further suggestion, and as indicated we reorganized Figure 5, regarding the result, we’d like to share with you that we're considering our experiments first of all as preliminary and then we'd like to specify that we performed that on particular and limited conditions as the cell culture in vitro assays, in the paper, we report the literature  investigation which collecting a large variety of papers that commonly debate miRNAs involving in adipogenesis but considering a lot of different conditions and models, so we had just tried to lead back our experimental result to literature but our next step will be deeply investigate and test biological mechanisms modulating those miRNA raised from this first step

While revising the manuscript another question came to my mind. While the authors, report that the different culture conditions of the two investigated adipose cells seem to affect the miRNA expression (results table 2), they did not comment on potential effects on the differentiation and maturation process itself. In the materials and methods section, it is described that differentiation was assessed by daily observation and the AdipoRed TM assay, a statement on the results of that for the two cell sources would be recommended. If there were differences in differentiation efficiency the impact of the different miRNA patterns might be discussed.

We thank the reviewer for the comment, which offers important food for thought on the models used for differentiation, and the differential miRNAs that characterize ASphCs and ADSCs. However, in the experimental setting used for this research, we did not constantly monitor the degree or timing of adipogenic differentiation of the two populations (something we observed in our previous work, doi: 10.4172/2325-9620.1000124). Therefore, the correlation of the specific miRNAs of the two populations with their adipogenic potential cannot be validated. However, this point will be addressed in our next study.

Finally, there is still some lack of information. In the section 2.2 (page 8) several references are missing, just as some examples line 254, 257, and 260. Moreover, there are still missing details on the patients such as age and gender and most of all number. What does “At least two replicates were run for each sample” mean for the miRNA Gene expression analysis? “Samples” referring to investigated groups (ASphCs, ADSCs, SDA and ADA) and “replicates” to actual sample size or is there more to it? In case of two patients, how much do the individual results vary?

We agree with the reviewer about the lack of statistical information about the groups size of each experiment. In particular, all the selected experiments were performed in triplicates.

Reviewer 2 Report

Comments and Suggestions for Authors

Comments on the revised version of the manuscript.

1) Lines 9-29:

The abstract (200 words maximum) should contain (without headings) short information about background (the question addressed in a broad context and highlighting the purpose of the study); methods in brief; summary of results; conclusion. Please follow the author’s instructions described in

https://www.mdpi.com/journal/ncrna/instructions

2) References: must be numbered in order of appearance (NOT ALPHABETICALLY!) in the text (including table captions and figure legends) and listed individually at the end of the manuscript.

Examples of correct reference presentation:

20. Berthet, E.; Chen, C.; Butcher, K.; Schneider, R.A.; Alliston, T.; Amirtharajah, M. Smad3 Binds Scleraxis and 447 Mohawk and Regulates Tendon Matrix Organization. J Orthopaed Res 2013, 31, 1475 1483, doi:10.1002/jor.22382. 448

21. Liu, H.; Zhang, C.; Zhu, S.; Lu, P.; Zhu, T.; Gong, X.; Zhang, Z.; Hu, J.; Yin, Z.; Heng, B.C.; et al. Mohawk Promotes 449 the Tenogenesis of Mesenchymal Stem Cells Through Activation of the TGFβ Signaling Pathway. STEM CELLS 2015, 33, 450 443–455, doi:10.1002/stem.1866.

Please correct your reference list in accordance with author’s instructions. In addition, put the authors' surnames before their first names in the links!

Throughout the text, reference numbers should be placed in square brackets [ ], and placed before the punctuation; for example [1], [1–3] or [1,3]. (PLEASE REMEMBER: NUMBERS, NOT SURNAMES!) For embedded citations in the text with pagination, use both parentheses and brackets to indicate the reference number and page numbers; for example [5] (p. 10). or [6] (pp. 101–105).

Please, change the references cited in tables 1 and 2 in the same manner.

Lines 155, 158, 160, 164, 165, 171: please insert square brackets with number of reference after surname: Sun et al [23].

3) Lines 70-72: obviously, this figure is deleted from the text

Introduction: it cannot be finalized by table (the table 1) but should contain, in brief, the information about the purpose of the paper, i.e. what purpose the authors pursued in their work.

4) The discussion is written too briefly. The authors should display the concept based on the data obtained and the limitations of the study conducted. Fig. 5 deserves more discussion based on the authors data.

5) Line 226: EGF-L – what is it?

Comments on the Quality of English Language

Minor improvement is deasirable

Author Response

Reviewer2

1) Lines 9-29:

The abstract (200 words maximum) should contain (without headings) short information about background (the question addressed in a broad context and highlighting the purpose of the study); methods in brief; summary of results; conclusion. Please follow the author’s instructions described in

https://www.mdpi.com/journal/ncrna/instructions

We thank the reviewer for having noticed this discrepancy in the abstract structure. We adapted the existing abstract following the guidelines of mdpi journal website, as follow:

Abstract: Mesenchymal stem cells due to their multipotent ability are considered one of the best candidates to be used in regenerative medicine. To date, the most used source is rep-resented by the bone marrow, despite the limited number of cells and the painful/invasive procedure for the collection. Therefore, the scientific community investigated many alter-native sources for the collection of mesenchymal stem cells, with the adipose tissue representing the best option, given the abundance of mesenchymal stem cells, and an easy access. Although the adipose mesenchymal stem cells have been recently investigated for their multipotency, the molecular mechanisms underlying their adipogenic potential are still unclear. In this scenario, this communication is aimed at defining the role of miRNAs in adipogenic potential of adipose-derived mesenchymal stem cells, by real-time PCR. Even if preliminary, our data show that cell culture conditions affect the expression of specific miRNA involved in the adipogenic potential of mesenchymal stem cells. The in vitro/in vivo validation of these results could pave the way for novel therapeutic strategies in the field of regenerative medicine. In conclusion, our research highlighted how specific cell culture conditions can modulate the adipogenic potential of adipose mesenchymal stem cells, through the regulation of specific miRNAs.

2) References: must be numbered in order of appearance (NOT ALPHABETICALLY!) in the text (including table captions and figure legends) and listed individually at the end of the manuscript.

 Examples of correct reference presentation:

Journal Articles:
1. Author 1, A.B.; Author 2, C.D. Title of the article. Abbreviated Journal Name YearVolume, page range

  1. Berthet, E.; Chen, C.; Butcher, K.; Schneider, R.A.; Alliston, T.; Amirtharajah, M. Smad3 Binds Scleraxis and 447 Mohawk and Regulates Tendon Matrix Organization. J Orthopaed Res 201331, 1475 1483, doi:10.1002/jor.22382. 448
  2. Liu, H.; Zhang, C.; Zhu, S.; Lu, P.; Zhu, T.; Gong, X.; Zhang, Z.; Hu, J.; Yin, Z.; Heng, B.C.; et al. Mohawk Promotes 449 the Tenogenesis of Mesenchymal Stem Cells Through Activation of the TGFβSignaling Pathway. STEM CELLS 201533, 450 443–455, doi:10.1002/stem.1866.

Please correct your reference list in accordance with author’s instructions. In addition, put the authors' surnames before their first names in the links!

Throughout the text, reference numbers should be placed in square brackets [ ], and placed before the punctuation; for example [1], [1–3] or [1,3]. (PLEASE REMEMBER: NUMBERS, NOT SURNAMES!) For embedded citations in the text with pagination, use both parentheses and brackets to indicate the reference number and page numbers; for example [5] (p. 10). or [6] (pp. 101–105).

Please, change the references cited in tables 1 and 2 in the same manner.

Lines 155, 158, 160, 164, 165, 171: please insert square brackets with number of reference after surname: Sun et al [23]

We would like to thank the reviewer for this suggestion about the references. Accordingly, we revised and corrected all the references-related issues.

3) Lines 70-72: obviously, this figure is deleted from the text

Introduction: it cannot be finalized by table (the table 1) but should contain, in brief, the information about the purpose of the paper, i.e. what purpose the authors pursued in their work.

The new Table 1 was indeed requested by Reviewer 1, with the aim to have a schematic description of all the miRNAs reported in the Introduction section, to make their role and involvement in adipogenic differentiation clearer to the readers.

4) The discussion is written too briefly. The authors should display the concept based on the data obtained and the limitations of the study conducted. Fig. 5 deserves more discussion based on the authors data.

Thank you for the observation, we revised the discussion as follow:

As described by many scientific works, adipose tissue is often used in regenerative medicine as an autologous filler for the correction of morpho-functional defects. In partic-ular, adipose-derived mesenchymal stem cells are increasingly recognized for their poten-tial in regenerative medicine due to several key advantages. Unlike bone marrow, which has traditionally been the primary source of mesenchymal stem cells but involves a pain-ful and invasive collection process with limited cell yield, adipose tissue offers a more abundant and easily accessible source of these cells. Adipose-derived mesenchymal stem cells have demonstrated significant promise in various therapeutic applications, owing to their ability to differentiate into multiple cell types, including bone, cartilage, and fat cells.

Thus, a better understanding of the molecular mechanisms underlying adipogenic potential of mesenchymal stem cells could help in the optimization of collection, culture, and treatment of this cell subset to improve their regenerative capacity. In this scenario, microRNAs (miRNAs) can be considered as crucial drivers of adipogenic commitment of adipose-derived mesenchymal stem cells, and at the same time as promising targets, to in-crease the clinical impact of regenerative medicine procedure based on the use of this cell subset.

Our data, obtained from this vast miRNAs’ analysis on adipose stem cell subpopula-tions together with their differentiated progenies, suggest that adipogenesis is carefully controlled by several factors (passage of cells, cell culture medium, adhesion versus ul-tra-low adhesion) and at many different levels. All these variables directly affect the ex-pression of specific miRNAs involved in the adipogenic potential of adipose-derived mesenchymal stem cells. We can speculate on a possible viable engineering therapy where an ectopic expression of miRNAs could be exploited to promote a specific commitment (i.e. we could get a pre-adipocyte population compared to a population of mature adipocytes, overexpressing mir27a or the mir31, limiting the harmful effect of adipocytes in lipofilling in cancer patients subjected to demolition surgery). Alternatively, we could try to overex-press mir100 or mir146a, succeeding in limiting the onset of breast cancer or succeeding in reverting a more aggressive phenotype of breast cancer to one that responds to anti-cancer therapies. Moreover, in the case of ectopic mir455, we could induce an adipose-type switch from WAT to BAT and thus minimize the damaging effects of obesity and harmful white adipose tissue paracrine secretion [47-48]. Moreover, another strategy could rely on the use of microRNAs identified as a possible strategy for the modulation of adipocyte-based pro-cesses. The delivery of such microRNAs through nanovesicles could deepen the study and mechanistic understanding of the biological and differentiation processes shown.

In sum, even if based on preliminary transcriptomic data, our results, if validated at pre-clinical level, by performing in vitro/in vivo analyses, could help to better define the optimal culture conditions and the main molecular driver of adipogenic differentiation, with important clinical impact in regenerative medicine.

5) Line 226: EGF-L – what is it?

We thank the reviewers for this correction, we fixed this issue.

Round 3

Reviewer 2 Report

Comments and Suggestions for Authors

The authors have done sufficient work to improve the quality of their manuscript. The reviewer accepts their revisions and wishes them further success.